# What about Using Photovoice for Health and Safety?

**DOI:** 10.3390/ijerph182211985

**Published:** 2021-11-15

**Authors:** Paul Lindhout, Truus Teunissen, Genserik Reniers

**Affiliations:** 1Faculty TPM-VTI (S3G), Delft University of Technology, Jaffalaan 5, 2628 BX Delft, The Netherlands; linteu@paal39.nl; 2Department of Medical Humanities, Amsterdam UMC, Location VUmc, De Boelelaan 1089a, 1081 HV Amsterdam, The Netherlands; truusteunissen39@gmail.com; 3Center for Corporate Sustainability (CEDON), KULeuven, Campus Brussels, 1000 Brussels, Belgium; 4Faculty of Applied Economic Sciences and Engineering Mgmt (ENM), University of Antwerp, 2000 Antwerp, Belgium

**Keywords:** health, safety, photovoice, scoping review, safety management

## Abstract

The positive reception of Wang and Burris’ photovoice method, published in 1997, has led to a proliferation of ways in which professionals deploy photovoice in a widening range of application fields, e.g., public health, social development and phenomenological research of human experiences. A scoping review method is used to obtain an overview of current photovoice designs and of application examples in the health and safety domain. The results show a variety of method designs. Our findings indicate that all of the photovoice designs are composed from different combinations of eleven process steps. Five generic objectives cover the range of application examples found in our literature study. We therefore condensed the variety into five generic photovoice designs for: (a) communication, (b) education, (c) exploration, (d) awareness, and (e) empowerment purposes. We propose this for use in a classification system. The potential for application of these photovoice designs in safety management is illustrated by the existence of various safety related application examples. We argue that the five generic designs will facilitate the implementation and usage of photovoice as a tool. We recommend that both a theoretical framework and guidance are further developed. We conclude that photovoice holds potential for application in health and safety management.

## 1. Introduction

The saying “*a picture can tell more than a thousand words*” was proven right many times since the rise of photography at the end of the 19th century. Photographs started to show what goes on outside peoples’ field of view in daily life, e.g., in countries overseas, in war zones or in remote places. Associated with each picture there is a story—a narrative or an explanation—told by the photographer or observer, sharing the experience on site [1,2]). Many journalists, scientists, and sociologists have used this basic concept to shed light on things that were hitherto unsaid, unknown or unresolved in society. While entering an increasing number of areas in society, a plethora of methods connecting image and story were developed over the 20th century [3]. During the 1980s participants were invited to make pictures about their life, and were then interviewed about these, thus leading to a ‘photo novella’ [4]. Several ‘participatory visual methods’ emerged, e.g., ‘participatory photography’, ‘photo-elicitation’ and ‘digital storytelling’ [5].

Currently, ‘photovoice’ is an increasingly popular method [6,7], which was originally developed in the nineteen nineties by Caroline C. Wang and May Ann Burris [4,8], and later refined [9,10]. Initially Wang and Burris (1997, p 369) [8] defined ‘photovoice’ as “*a process by which people can identify, represent and enhance their community through a specific photographic technique*”. At the time, photovoice was designed to reinforce empowerment among those who live in poverty or marginal situations and to generate collective action based on the exhibition of photographs voicing pressing issues to leaders and policymakers [9]. 

Present use of photovoice is no longer constrained by the term ‘community’ since, e.g., ‘individual experiences’ are being included in many studies in recent years [11]. Currently, ‘action’ is not always part of a photovoice project design [6]. 

As a method, photovoice offers modes of expression to people in addition to a mere verbal way of sharing their experiences [12]. This helps in groups with, e.g., limited verbal skills, low proficiency in the local language [13] or with complicated and emotional experiences in their lifeworld [14]. This inclusive and action-oriented design was based on the notion that photo-stories can be used on a collective level to achieve changes in policy, quality of life and care in the neighbourhood, city and society [15,16,17,18,19,20]. This diverse usage extends beyond the original principles of photovoice. Although an increasing variety in photovoice applications is observed [7], there is no classification and only little guidance for choosing the most suitable photovoice design for a given problem setting [21]. Today’s photovoice applications are no longer always grounded in Community Based Action Research [4,8]. In their application status review, published in 2010, Catalani and Minkler [6] found that at that moment only 60% of photovoice projects included an ‘action’ component. Although sources often refer to the original 1997 theoretical framework of Wang and Burris [8], practitioners and researchers currently vary their application of photovoice according to their own insight [1] and no longer always include empowerment as their primary objective. 

This gives rise to several methodological, practical- and ethical issues. Current safety management practice is increasingly often confronted with both the human aspect, e.g., safety culture, behaviour based safety, worker group interactions, and with more complex installations, processes and environmental constraints. Safety management may benefit from using methods which provide more insight in what is happening between, around and to people interacting with each other, with equipment on the shop floor, and with their work places, e.g., in industry and in health care [13,22,23]. Furthermore, both safety management and workers could benefit from the application of methods which can be used by the people themselves [24,25]. Photovoice would be an effective and powerful tool, e.g., for workers in a specific setting, for workers represented by a union or for communication with safety regulatory agencies. Since the use in the health and safety management domain is rather limited, the photovoice method would be worth considering in this respect. Because of its persistent growth and a substantial existing application experience in a wide range of application areas in society, the photovoice method may be expected to hold significant potential for further expansion of its use in safety management. Where in their 2010 review, Catalani and Minkler [6] focused on the process and outcome of photovoice studies and projects, we seek to gain insight in both the emerging variety of photovoice designs and in the possibilities for their application in health and safety management. Hence, we conduct this study in search of an answer to the following question:


*What can be said about the design variety observed in current photovoice application practice and how can this method be applied in health and safety management?*


## 2. Materials and Methods

Since the photovoice method originates from the 1990s [8], and has been reviewed in 2010 [6], we focus on the method’s current state. To this end we use a time period starting from 2010 until 2020. By July 2021, a Google search on the term ‘photovoice’ generated some 838,000 hits, whereas the same search in 2019 had resulted in 563,000 hits. This indicates a considerable growth in usage. A look into the content of this high number of hits makes it immediately clear that the photovoice method is not only being increasingly used, but is also spreading out over a wide range of societal problem areas in many countries. It also indicates the need for a clear focus and pre-set limitations to our present study. 

Inspired by the work of Catalani and Minkler (2010) [6], Coemans et al. (2015) [26] and Lal et al. (2012) [7], we use the scoping review approach [27], and a literature screening process [28,29] with search terms and several inclusion/exclusion criteria. 

Accordingly, in order to be admitted, sources must be primary, peer-reviewed scientific sources, with content pertaining to the photovoice methodology, its origin, development, current application in practice and its theoretical framework. We excluded sources published before 2010 or dealing with non-Western societies. By exception, a few older sources were included because of their particular relevance for the history of visual methods or because they reflect well documented practice examples of photovoice usage. We included several relevant non-academic secondary sources [30] and credible tertiary-often called “grey”-sources [31]. We used Google Scholar and the associated ResearchGate-, Academia- and proprietary publisher repositories as general databases. 

Our search strategy is aimed at finding sources about photovoice methods and application. To this end we performed a preliminary search, a method oriented general applications search and extra searches focusing on health and safety related photovoice applications.

A-Preliminary search.

A set of currently used terms is derived and progressively refined during successive searches to find the origin of photovoice as a method and obtain currently used terms, apply different combinations of currently used terms and to establish inclusion/exclusion criteria. Then a suitable set of search terms is derived from these preliminary findings. In this study the terms are: Photovoice method AND social community AND participatory action research AND lived experiences AND personal life world AND collective action AND low socioeconomic status AND ethics AND empowerment AND unarticulated OR nonverbal OR unspoken OR unsayable.

B-Method oriented general search

Next, an online search with these search terms is conducted, using inclusion/exclusion criteria to decide on the admission of sources found. The reference listings of admissible sources were then screened for further admissible sources in a separate search. Finally, a manual search in sources available to the authors completed the method oriented search.

C-Extra searches 

In order to find specific health and safety related photovoice design application examples, three extra searches are conducted, using “occupational AND health AND safety”, “patient AND safety” and “health AND promotion” as additional search terms. 

## 3. Results

### 3.1. Literature Search Results

The method oriented literature search yielded 161 sources in total, of which 146 were primary scientific sources and 15 were other sources. The extra searches yielded 69 sources, most of which were primary sources, see Table 1. 

An overview of the literature search and selection process is presented in Figure 1.

### 3.2. Method Oriented Search

The contents of the method oriented sources led to several observations on the design variety of the many photovoice applications, e.g., in social justice [32], in needs assessment, in empowerment of groups of people in society, in health care, in therapy, in education and in public health research. We screened these sources to shed light on the theoretical framework, on the process, on available guidance and on additional techniques used in photovoice practice.

### 3.3. Theoretical Framework

Different points of view exist with respect to the theoretical reference frame of photovoice. Three reference fields are mentioned [33,34,35]: empowerment education, feminist theory and documentary photography. The term ‘photovoice’ underlines the necessity of both the visual part—the photographs—and its explanation by the maker, the narrative part [36]. In the form of counter stories, a narrative is a means to criticize and change the dominant discourse in society [37]. Some sources mention a role for ethnography and visual anthropology [38], also named: visual ethnography [39]. This would imply that, besides Community Based Participatory Research (CBPR) [4,6,8,40,41,42] and Participatory Action Research (PAR) [43,44,45,46,47], there are even more theoretical reference frames for photovoice, namely: Arts Based Research (ABR) [10,48], and more specific: Visual Research [2,5,21,27,49,50,51,52,53,54,55,56,57] and Narrative Inquiry [58,59].

### 3.4. Process Steps

Most current applications of photovoice make use of a limited number of process steps, as first outlined by Wang and Burris in 1997 [8]. Since then, the number of steps varies in literature. Wang and Burris (1997) [8] and Wang (2006) [60] use nine steps. Plunkett et al. (2013) [3] divide the process over six steps. Liebenberg (2018) [61] describes three steps, Abma, Banks, et al. (2019) [62] use ten steps. In fact, all these sources combine the same activities into more or less process steps. Although some minor divergence is observed in how steps are carried out [47], their sequence aligns well with the photovoice process as originally designed. 

We observe that fully fledged photovoice processes in practice can be broken down into eleven separate activities in a logical order. We therefore chose eleven steps to describe the process in its entirety. Merging of process descriptions from many sources allows a generally applicable description per photovoice process step. For each step, we added references to various literature sources, presenting generally applicable information about relevant theoretical background and use in practice, see Table 2.

### 3.5. Techniques

Over the range of designs, several methods and techniques appear in photovoice practice. We allocated these to the process step #7-Group sessions in Table 2 and included references there.

### 3.6. Guidance

Practice shows that many photovoice applications do not make use of all eleven steps or techniques. However, no generally applicable criteria for inclusion of any specific step or technique in a photovoice design were found in this study. The same applies to photovoice as such, e.g., in a range of other visual methods. Both overlaps and differences between photovoice and various other methods using photographs are found and it is pointed out that all of these are to be seen as in support of the narrative result of a project [61]. No guidance was found on choosing a photovoice method for a given research question or combining this with other ABR methods or verbal research methods, although there is a clear need for it [48,66].

In practice however, only limited guidance is available for photovoice application in specific areas, such as personal development, advocacy, public consultation, social research, health research, education, disability studies, collaborating with youth in school, and finding issues and needs [64,65,66,67,94,95].

### 3.7. Extra Searches

Since no preclusions or obstacles were discovered, we set out to explore the potential of photovoice in the health and safety management field. This not only requires a clear definition of the characteristics of the photovoice method as such, but it also needs inspiring examples from practice. To this end we explore three areas, relatively close to safety management, in more depth.

### 3.8. Occupational Health and Safety Management Applications

Similar to applications in other fields, not all of the photovoice applications in health and safety management will require an action component [6]. Photovoice is applied, e.g., as an educational tool [96,97], for investigation of environmental pollution and health impact [98,99,100], to support workplace hazard identification jointly with workers [24,25,101,102], to examine healthcare workers work related injuries [103] or disparity of minority groups incident rates [104,105,106], to assess social factors for workers in relation to occupational disease [107] and to increase awareness of factors affecting mental health at work [108].

Photovoice application examples from occupational safety practice exist on the objectives of hazard identification, education and training, investigation of environmental impact, examining causality of occupational disease/injuries, gaining understanding of incident rate disparities, increasing awareness and assessing social and mental factors affecting workers’ safety.

### 3.9. Health Promotion Applications

A wide variety of photovoice application design examples is found in public health improvement activities [6,7,15,16,19,20,53,75,83,109,110,111,112]. Other health promotion examples deal with, e.g., accurate representation of a group viewpoint [113], improving family relations and stimulation of healthier behaviour [114], teaching at school [94,115,116], train the trainers to be facilitators rather than experts [111], exploring homelessness [117], the situation of refugees [118], sex and drugs related behaviour [119,120], assessing needs of marginal groups in society [110,121], better understanding of a group experiencing problems, e.g., in coping with a disease [112,122,123], with the troubles of adolescence [75,124], with access to local bus-stops [125] or with poor housing [126].

Photovoice examples from the health promotion field exist on health related behaviour improvement, on using education tools, on exploring what goes on in a group of people with a specific problem and on assessing the needs of a group of people in a specific setting.

### 3.10. Patient Safety Applications

Photovoice is often used in the social domain and has entered the health care and well-being fields, including patient safety. Patient safety requires closer attention and the most effective way to achieve this is not only evaluating the incidents occurred, but also look at near-misses since these happen some 300 times more often [127].

A search for scientific sources shows that photovoice is being used in relation to well-being and safety of patients with a variety of both physical health problems [128], e.g., diabetes [68,72], asthma [129], heart disease [130], and mental health problems, e.g., violation victims [131], war induced PTSS syndrome [132], mental illness or experiences [133].

Several scientific studies address physical patient safety, e.g., in relation to fall prevention for wheelchair users [85], transfer and lifting of patients [134], and connection with equipment via catheter, for intravenous medication or for breathing with medical oxygen [103]; however, many other patient issues can touch upon safety and well-being [135]. Several example applications addressing improvement of patient care and affecting safety and well-being were found [6]. Examples of unsafety are, e.g., caused by illiteracy among patients and not understanding of the local language [23].

Several so called “grey” sources [31] show that photovoice is being used in various ways for patient safety improvement purposes, e.g., in relation to Intensive Care Unit room design [136], daily care in hospitals [137], suicide risk [138], drug abuse [139], stigma [140], safety perception of mentally impaired youth [141]), people with misunderstood behaviour [142,143] and people with dementia [144]. In mental care situations the role of family and friends must be included since patients are not always able to make pictures and express themselves verbally [145].

Photovoice is being used with diverse objectives. It is, e.g., being used as a training system tool for nursing education [74], to investigate self-care among homeless diabetes patients, focusing on needs assessment of this group [68], to explore health inequities in minority groups [146], and to investigate conditions in urban areas with a high diabetes incidence [72] where researchers found that their perceived availability of food resources to patients does not match with reality, leading to recommendation of community action to increase awareness. A review of photovoice method applications use, conducted in mental health research, shows that exploring a situation (needs, preferences, experiences) within a group often is the main objective [128]. Hospitals have implemented safety management systems, but patients could contribute more actively to improve patient safety, e.g., by error prevention [147]. This could be done, e.g., via spotting near-miss situations [137].

Photovoice application examples from the patient safety field include both physical and mental safety aspects and employ project objective based variant designs for a.o. training, increasing awareness, needs assessment, exploring situations in specific groups, and improvement of health care and patient safety. In addition to the patients, their family and friends may also need to be involved.

## 4. Analysis

### 4.1. Photovoice Project Objectives

The examples from practice, in the above mentioned three areas close to safety management, appear not to differ significantly from the photovoice examples found in other societal areas. This suggests that the common process steps listed in Table 2 can also be used as a basis for applications in health and safety management. Clearly this would also need further development of guidance [21], e.g., starting from the sources referenced in Table 2.

We observe that the differences between photovoice designs from all application areas appear to be not so much in the depth and rigor with which each of the process steps are executed, but rather in whether or not a specific process step is included in a photovoice project design.

This implies that a single specific photovoice design or model for all health and safety management applications does not seem feasible. In turn, this would mean that the full variety of photovoice designs found in practice is also applicable to safety management. In order to avoid complicated theoretical and practical preparations per application case, and because the possible number of combinations of the eleven process steps is very large, we used clustering to find a small number of frequently used process step combinations. This opens the more practical possibility to use several generic photovoice project designs instead of unique designs for each project.

To this end we screened the literature sources describing application projects for details about where in society these were conducted, what the objectives were, which of the eleven process steps were applied, and what kind of outcomes were reported. Our findings indicate that the differences between photovoice process designs are strongly linked to project objectives and much less to locations, situations or settings where projects are conducted.

### 4.2. Generic Photovoice Designs

Using simple text analysis and meta-synthesis [148] we were able to allocate all of the examples found from current photovoice practice in our literature search to five generic objectives. These generic objectives provide the basis for five generic photovoice designs, which we named A, B, C, D and E; each of these was built from a number of steps, selected from the eleven process steps. We illustrate these generic photovoice designs and their building blocks in Figure 2, listing them in order of increasing number of included steps; e.g., the A-Communication objective requires three–nr 2, 3 and 4–process steps, whereas the E-Empowerment objective requires all eleven steps nr 1–11. In support of the choice for one of the five generic designs, considered as best suitable for use in a given setting, a description per generic photovoice design is provided below. This description explains why the selected process steps are used in a wide range of applications in society. Several examples underline the potential for use in health safety management.

#### 4.2.1. A-Communication

The Communication oriented design is most frequently used as a tool in professional applications [7]. In this design, the photovoice process is limited to preparation, photoshoot and presentation steps. Photography is used to communicate participants’ experiences among each other and to external change agents. The use of photography is pragmatic and focuses on the impact these photographs have on external others. This most compact photovoice design is used in a wide variety of practical applications, both by the general public and by health care professionals, social workers and researchers [7]. Professional use provides, e.g., a means of communication within a multi-method approach [149] or supports the data gathering activities in practice oriented ethnographic research [112] or in public health improvement projects [113,114]. In health and safety management this generic photovoice design for communication could be used, e.g., to methodically share observations of equipment degradation, pollution sources, accident causality, types of injury and other visual observations [100,103,104].

#### 4.2.2. B-Education

In support of Education a similar tool with some planning and preparation before the photo shoot step and discussion during the presentation step is being used as an interactive teaching tool, e.g., for projects in classrooms [64,89,109,150,151,152]. The goals with this design are to educate the participants themselves, to foster a reflective process among participants, to obtain new insights in learning and to keep participants actively engaged [94,109,115,116]. In health and safety this generic photovoice design for education, may be used as a tool to assist, e.g., safety education and training in an interactive and practice oriented way [74,96,97].

#### 4.2.3. C-Exploration

The Exploration of a situation requires the process steps from planning up to and including data analysis. An exploratory study is often conducted to assess the—as yet unknown—needs of a group [8]. An interviews step is included in the design. We chose “exploration” since it includes not only needs assessment [8] but also gaining understanding of people’s culture, situation or specific behaviour. In such projects, neither empowerment nor social action steps are planned beforehand, since the study outcome cannot be sufficiently predicted. Examples of this are finding a cultural identity [20], obtaining insight in the lifeworld of a specific group [153,154,155,156,157,158,159] or gaining understanding about, e.g., health, food and sports, abuse or drugs related behaviour within a group [16,49,110,112,117,118,119,120,121,160,161,162,163,164,165,166,167,168,169,170,171]. Health and safety risk assessment can be assisted by using this generic photovoice design for exploration, e.g., of a work environment such as in mining, construction, waste handling, etc., occupational hazards in a workplace, social factors, a neighbourhood, or challenges experienced by foreign workers [24,25,85,99,101,102].

#### 4.2.4. D-Awareness

Creating Awareness in participants’ private situations at home, at work, in a cultural group or in society is often done with individual or group empowerment in mind. Applications are, e.g., used to recognize a common problem within a group. This requires the photovoice process steps to continue up to and including group sessions and dissemination of the findings. The result will often be a recommendation for future action [9,16,172,173,174,175]. Various examples address the increase of common collective awareness of e.g., social, medical or health problems [75,122,123,124]. Sufficient awareness of the hazards at a workplace is of paramount importance for workers’ occupational safety, for occupational disease risk, and for company safety management. Application of the generic photovoice design for awareness can also be used, e.g., to identify undesirable effects of routine behaviour [100,107,108].

#### 4.2.5. E-Empowerment

Finally, there is the Empowerment design, closely following the original design of the photovoice method [8], which aims at realizing collective empowerment of a group of people. Further process steps for Action, Societal change and Evaluation [11] are included in the process design after dissemination of the findings. Here it is all about social problems, e.g., in a neighbourhood, cultural—or ethnic community, and taking action for improvements with and to the benefit of the entire community [17,34,39,40,63,75,92,110,111,126,176,177,178,179,180,181,182,183,184,185,186,187,188]. Safety improvement relies on each of the workers embracing safe conduct instructions and reporting unsafe situations and near misses. Workers can deviate from this line for various reasons though. This generic photovoice design for empowerment can help to achieve their active contribution in a methodical way while workers are supported by company management and have no hesitation, e.g., due to job insecurity [105,106,189].

## 5. Discussion

Companies and institutions with their own health and safety management often work top-down and support their safety related activities with analysis, records and expert reports. Some companies engage in behaviour based safety activities. Fewer companies and institutions engage in bottom-up activities in support of safety management.

### 5.1. Potential for Safety Management

Photovoice has not yet reached its full potential and has few restrictions, although, e.g., budget, job insecurity or illiteracy must be addressed [190]. Wang and Burris (1997) [8] argued that “*photovoice . . . uses the immediacy of the visual image to furnish evidence and . . . a . . . participatory means of sharing expertise and knowledge*”. Reflecting on the existing health and safety related photovoice applications found in this study and the similarities with what safety management system activities need to accomplish and improve in organizations [22], we observe that:

The main strength of photovoice is that the data are ‘pure’ since participants, in this case, e.g., workers in a factory, nurses in a hospital department or iron workers on a construction site, both make and interpret the photographs themselves. The data are also accurate since they represent what a group of people is experiencing in practice in settings which may be difficult to explore with other methods. Photovoice is a well proven tool with the strength to convince people in charge, producing results based on robust evidence from practice and with the support of people in the situation being examined. Photovoice as a tool is useful to safety management since it can withstand criticism and resistance from those who protect their interests at the expense of health and safety and it helps to create joint ownership of a project outcome among participants and stakeholders. Using the photovoice approach helps to visualize and to provide underpinning of observable safety behaviour, safety culture and safety climate. Photovoice is suitable for many purposes in education and training in many areas, hence it seems likely that this also applies to health and safety. 

The photovoice method, applied in health and safety management, can open up new access windows to hitherto hidden and unknown situations, like it does in many other areas in society. Given its successful track record in many areas in society, we contend that photovoice has a high potential as a tool in support of health and safety management.

### 5.2. Implementation Path

Introduction of photovoice as a new tool in health and safety management is possible without major financial, knowledge or manpower requirements. It can be done in six steps:Write a policy on where and how photovoice can be used as a method and determine criteria, conditions and limitations for use and/or distribution of photographs and notes;Issue a procedure outlining the five generic photovoice designs, providing work instructions on how to conduct each of the eleven steps and specifying what needs to be noted to allow proper data processing and storage (The information needed for this can be derived from Table 2 and Figure 2);Provide cameras (e.g., smart phones), a meeting room with a projection screen and/or display board, a storage facility for photographs and notes, and a printing facility;Decide on which photovoice generic design is best suitable for each new case and setting;Obtain written consent of participants as necessary at the start of each new project;Follow the agreed policy, procedure and instructions.

### 5.3. Ethical Issues

Several ethical issues need to be addressed in photovoice projects. The most frequently mentioned ethical issues with photovoice are: creation of unsafe conditions due to increased visibility of individuals, public exposure of participants, stigma, violations of confidentiality, invading privacy, involuntary intimacy, stereotyping, incriminating, framing, making profit on needy people, publishing of photographs of participants without their permission, disputing ownership of participants’ photographs, not ensuring informed consent, not taking into account illiteracy, not including an accurate expression of the group voice in reports [7,11,29,34,62,63,65,69,70,79,91,92,93,189,191,192]. Also transferring a specific design of the photovoice method to other settings requires attention for ethical aspects [63].

Protection of participants is an ethical concern in all projects [3,20,48,63,65,92,167,172,193].

If a photovoice project reveals hitherto unknown facts, leads to new awareness and insights, does not accomplish pre-set goals, shares more photographs or information than people want, or is not done according to expectations, people might want to reconsider their position or even their participation [73,194]. Photovoice is a dynamic process and may encounter unexpected problems. A way to adjust a running photovoice project on the basis of questions raised underway is needed [195,196]; although in response to this, several ways exist to reduce drop-out rates [109]. Participants, e.g., workers, patients, need to know they are being protected against losing, e.g., their job, reputation or social position.

### 5.4. Methodological Issues

In practice several problems are encountered, mainly the general issues: logistics, capacity, and accessibility [7]. Photovoice projects also suffer from poor method definition, namely, as applies to data recording, reporting and evaluation [6,27,197]. The sample size for photovoice projects is generally small. This means that the results within the group of participants cannot always be generalized to the entire population [181]; however, a choice for a larger group of participants, a longer time period and use of other arts based methods can add robustness to the findings [6,198,199]. We suspect that the wide variety in currently used photovoice application designs and the lack of guidance about when and how to apply a particular variant, might pose a threat to quality, robustness and credibility of photovoice project outcomes.

The rather complicated concept of empowerment is not always made clear for participants beforehand [200] or is not achieved, either intentionally or not [6,33]. A photograph’s meaning cannot be determined without an explanation by the maker [1,2,78,80,201]. Combined photographic and narrative data allow the researcher to view life experiences through the participant’s eyes. The narrative is an essential part of the photovoice method [36].

Photovoice project designs cannot be executed the same way for all groups and settings. A group of people might be limited in their activities by an illness or disability, an indigenous group might require consideration of cultural aspects [202] and poor language proficiency can affect the quality of the ‘narratives’ explaining participants’ photographs [36,162,203]. Interpretation of children’s voices can be complicated by the layered structure of expression and non-normative behaviour [56], by intrusion in their private space, hitting children’s limits and taboos [204] and by the challenges posed by the co-researcher role of children [205].

A unified set of terms in photovoice projects has not yet emerged [197] although steps towards a worldwide community of practice on photovoice are being taken [67,109].

### 5.5. Limitations

An overview of the photovoice method’s strengths and weaknesses was not found although several sources identify some of them [180,181]. The actual limitations of the photovoice method are therefore unclear [40,172,194]. The boundaries of photovoice can be disputed. This becomes clear from two extremes: merely taking photographs as data generated by an individual does not qualify as photovoice in our view. In this study, we do not consider photograph supported self-examination by patients, e.g., for a check on skin melanoma [206], or during online consulting by a general practitioner [207], as part of the photovoice method design range. Neither do we consider photographs or video footage, uncoordinatedly made by individual patients, e.g., included in articles, blogs or social media expressions, as a part of the photovoice range. The other end of the range, in our view, is where the media generate huge numbers of static and moving images with explanatory text, all influencing the development of society on national and even global scale.

### 5.6. Future Developments

Information and communication technology is increasingly used and leads to integration of static imagery with voice, video [208], social media [27,41], web-assisted ways to engage people [83], citizen science [209] and larger scale online applications [210]. This trend follows a general phenomenon: the rise of visual research methods [55].

## 6. Conclusions and Recommendations

Photovoice studies are increasingly found worthwhile because the methodology allows creating room for the voices of people living in poverty and marginal situations, and they offer insight in the lifeworld context of people not often heard from by decision-makers. Many current researchers and practitioners vary on the original photovoice design as outlined by Wang and Burris in 1997 [8].

We identified five generic photovoice designs covering the range of photovoice application objectives found in current practice: communication, education, exploration, awareness and empowerment. In order to avoid complicated theoretical and practical preparations for each project, we recommend choosing from these five generic photovoice designs. These need an explicit theoretical justification and the development of specific guidance. We contend this is important, also in support of safeguarding the quality of project outcomes.

On the basis of a proven track record of photovoice in several specific safety management related areas and a clear and growing presence in many other societal areas, we argue that photovoice has a high potential as a tool in support of health and safety management.

## Figures and Tables

**Figure 1 ijerph-18-11985-f001:**
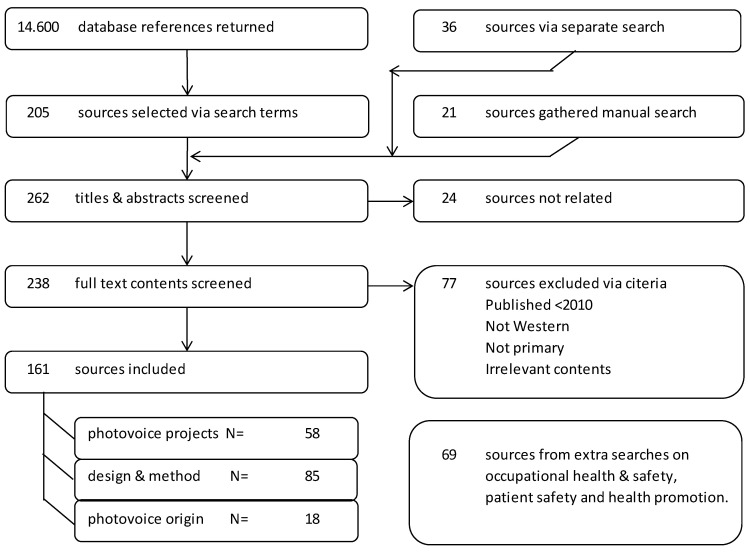
Flowchart of literature screening process [28,29].

**Figure 2 ijerph-18-11985-f002:**
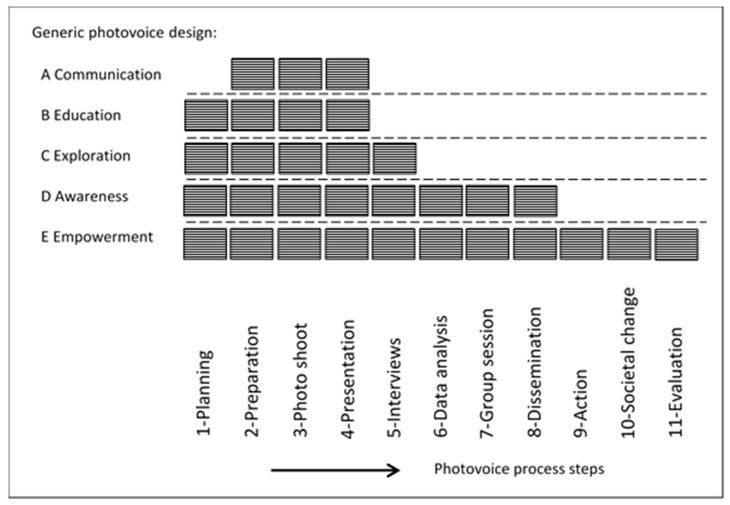
Five generic photovoice designs and their process steps.

**Table 1 ijerph-18-11985-t001:** Primary, secondary and tertiary sources found from literature and selected via criteria.

Source Type	Method Oriented Search	Extra Searches
Primary	146	52
Secondary	8	6
Tertiary	7	11
Total	161	69

**Table 2 ijerph-18-11985-t002:** Common process steps observed in photovoice projects with references.

Step Nr	Subject	Description and References
1-PLANNING	Getting started	Initiate and present a plan for social action, gather action group, set target audience of policy makers and community leaders, create partnership, set project goal and project planning [8,33,60,62].
	Preconditions	Presence of reflective experiences and a community requiring change [8,61,63].
2-PREPARATION	Set-up	Determine the research question, establish the project design, choice of participatory methods, define participant role (co-researcher, quality of participation), co-creation of research design [38,62,63,64].
	Resources	Gather materials (space, cameras, other resources, arrange for photo printing), arrange (financial) compensation, decide on support from translator, photographer, facilitate small and large group meetings [8,65,66,67]. Determine or establish a setting/group/environment [65].
	Recruitment	Selection and recruitment of facilitators and participants [8,38,60,68,69,70,71].
	Training	Introduce photovoice method to participants, facilitate a group discussion about cameras, power, and ethics. Distribute cameras, provide camera instruction. Discuss and manage Reality and Arranged scenes. Manage relevance, number and selection of photographs, photography of people (either avoid it or get consent), how to obtain informed consent [3,8,34,60,63,64,67,70,71,72,73,74].
3-PHOTO SHOOT	Assignment	Identify and discuss photo assignment, decide on initial themes for taking pictures [34,60,69,70].
	Fieldwork	Fieldwork (allocate time to take pictures, set themes, go out into the community, take pictures, take field notes in logbook (self-reflections, insights, record /contemplate aspects of the research; observations of community, climate, viability, health, photovoice); write down photo captions/titles [2,3,4,34,38,60,64,67,71,75,76,77].
	Data gathering	Collection, initial review and analysis of photographs and logbooks, follow-up [3,38].
4-PRESENTATION	Selection	Selecting photographs, context and story, meet and discuss issues, themes and theories [60,67].
5-INTERVIEWS	Share	Conduct individual photo-sharing sessions, decide what is data, establish meaning of photos [1,2,3,63,78,79].
	In-depth	Conduct individual in-depth participant interviews (value, feeling, knowledge, and sensorial) and analysis [3,71,76].
6-DATA ANALYSIS	Organize data	Making sense of conversations, stories, photos, performances, documents, data or non-data [62,72,80].
7-GROUP SESSION	Group review	Collective photo-sharing, analysis, selection, interpretation and discussion of photographs, stories, context, documents. Best picture choice, co-construction and codification of meaning, issues, themes, theories [3,8,14,15,19,61,62,64,69].
	Methods and techniques	LOOK [81], PHOTO [63,82], SHOWeD [14,15,19,60,62,68,83,84,85,86,87], VOICE [8,61], 4 STEPS [88], PA Participatory Analysis [3].
	Visual analysis	Perform Visual Analysis [6,14,61,63,84], consider EMIC (participant opinion) and ETIC (patterns) perspectives [38,65].
	Narrative analysis	Inductive thematic analysis, content analysis, narrative analysis [63,80,89].
	Decision	Need for another round: esthetics (e.g., need for remake of pictures, more data gathering and analysis), answering the research question, closure, debrief [33,62,64,65].
8-DISSEMINATION	Sharing	Plan a format to share photos/stories with policy makers/community leaders, exhibition, linking to social change, advocacy, generate impact [33,60,61,62,63,64,66,71,79,90].
9-ACTION	Audience	Select, recruit and reach target audience of policy makers/community leaders to create change [61,66,69].
10-SOCIETAL CHANGE	Partnership	Sustainable partnership, knowledge and outcomes [61,62,70].
11-EVALUATION	Project	Evaluation of project (success criteria, output/outcome, participation quality), conditions (exposure of participants, confidentiality, privacy, intimacy), ethics (stereotyping, incriminating, false light, profit, publish, informed consent), legal compliance and quality [11,29,34,62,63,65,69,70,79,91,92,93].

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
