# Peer review of "What about Using Photovoice for Health and Safety?"

_ijerph, 2021, doi:10.3390/ijerph182211985_

Round 1

Reviewer 1 Report

Peer Review of “What about using photovoice for health and safety?” for IJERPH

I appreciate the directness of the title.  For clarity it might be useful to add occupational to health and safety in the title and at some points in the paper so as to draw reader with that interest.

Abstract – Is complete and represents the study accurately.  In the sentence beginning “Only five generic objectives…”  “Only” could be removed for clarity, Line 18.

Introduction – Thorough description of photovoice context and development. Interesting and learned quite a bit although I’ve used photovoice. This should be a good intro for those who haven’t used it or are considering using it.  On Line 54, limited verbal skills would be a more standard usage.  Line 67 “no longer include empowerment always as their…”  “do not always include…” would be more clear.

Paragraph beginning line 69: The paper seems to be aimed at safety managers exclusively.  Photovoice would also be a powerful tool for workers represented by a union or other worker organization or health and safety regulatory agencies. Could you perhaps expand on those possibilities either here or in the discussion?

Methods – Search process and inclusion/exclusion criteria well described and consistent with table. 

Results - Table 2 was nicely set up, easily readable and useful for planning a potential project. On line 182 you might add that Group Session is #7 on the table. Readers might like to see paragraph 3.8 on Occ health and safety expanded somewhat and perhaps less on patient safety. On line 246: “unsafety” isn’t standard usage. Perhaps “unsafe conditions”?

Analysis - The one-sentence paragraph beginning on Line 280 could be worded more clearly.  Figure 1.:  The empowerment objective is missing.  Line 343 “forehand” should be “beforehand”.  

Discussion- Line 383 “little restrictions”, the more common usage would be “few restrictions”.  The observations beginning on Line 390 are convincing but the choice of bullet points to present these makes for some redundancy. This would work better as narrative. The bullet points are appropriate for the six implementation steps. 

Ethical issues paragraph 5.3 need to be expanded, particularly the first paragraph.  I was unsure what some of the references such as “false light” were referring to.  Again, it needs to be in narrative rather than bullet form.  What were strategies the projects referenced used to protect their participants.  Put more plainly:  if I was an industrial or health worker, I would want to know exactly how presenting a photo of a safety violation is not going to lead to my termination.

Limitations – there should be a few sentences more on your study limitations.

Future developments – Recommendations for choosing from the five generic designs is helpful for other researchers.  Overall, this paper is a good introduction for those considering photovoice as a research tool and is a meaningful contribution to the participatory research frameworks.

Author Response

REVIEWER 1 Comments (see authors response in blue )

Moderate English changes required
            We have included the improvement suggestions made by the reviewers.

. . .

I appreciate the directness of the title.  For clarity it might be useful to add occupational to health and safety in the title and at some points in the paper so as to draw reader with that interest.

            We have considered adding “occupational” but we did not adopt this suggestion since it would exclude other areas of safety activities which could benefit from photovoice. Our intention is to propose photovoice as a tool not exclusively in occupational safety but also e.g. in Seveso III related major hazard control, in health care institutions (patient) safety, in public safety, and in other health and safety related activities.

Abstract – Is complete and represents the study accurately.  In the sentence beginning “Only five generic objectives…”  “Only” could be removed for clarity, Line 18.

            We have included the improvement suggestions made by the reviewer.

Introduction – Thorough description of photovoice context and development. Interesting and learned quite a bit although I’ve used photovoice. This should be a good intro for those who haven’t used it or are considering using it.  On Line 54, limited verbal skills would be a more standard usage.  Line 67 “no longer include empowerment always as their…”  “do not always include…” would be more clear.

            We have included the improvement suggestions made by the reviewer.

Paragraph beginning line 69: The paper seems to be aimed at safety managers exclusively.  Photovoice would also be a powerful tool for workers represented by a union or other worker organization or health and safety regulatory agencies. Could you perhaps expand on those possibilities either here or in the discussion?

            We agree with this comment. We have expanded the sentence to: “Furthermore, both safety management and workers could benefit from the application of methods which can be used by the people themselves [24,25]. Photovoice would be an effective and powerful tool e.g. for workers in a specific setting, for workers represented by a union or for communication with safety regulatory agencies.”

Methods – Search process and inclusion/exclusion criteria well described and consistent with table. 

            Thank you for this comment.

Results - Table 2 was nicely set up, easily readable and useful for planning a potential project. On line 182 you might add that Group Session is #7 on the table.

            We have included the improvement suggestions made by the reviewer.

Readers might like to see paragraph 3.8 on Occ health and safety expanded somewhat and perhaps less on patient safety.

As explained  in the above, our intention is to propose photovoice as a tool not exclusively in occupational safety but also e.g. in Seveso III related major hazard control, in health care institutions (patient) safety, in public safety, and in other health and safety related activities. We feel that the chapters 3.8, 3.9 and 3.10 are proportionate.

On line 246: “unsafety” isn’t standard usage. Perhaps “unsafe conditions”?

            We have included the improvement suggestions made by the reviewer.

Analysis - The one-sentence paragraph beginning on Line 280 could be worded more clearly.  

We changed the sentence to: “Using simple text analysis and meta-synthesis [148] we were able to allocate all of the examples found from current photovoice practice in our literature search to five generic objectives.”

Figure 1.: The empowerment objective is missing.  

            Thank you for spotting this graphical flaw.

Line 343 “forehand” should be “beforehand”.  

            We have included the improvement suggestions made by the reviewer.

Discussion- Line 383 “little restrictions”, the more common usage would be “few restrictions”.  

            We have included the improvement suggestions made by the reviewer.

The observations beginning on Line 390 are convincing but the choice of bullet points to present these makes for some redundancy. This would work better as narrative.

We concur with this comment and have changed the text accordingly:

“The main strength of photovoice is that the data are ‘pure’ since participants, in this case e.g. workers in a factory, nurses in a hospital department or iron workers on a construction site, both make and interpret the photographs themselves. The data are also accurate since they represent what a group of people is experiencing in practice in settings which may be difficult to explore with other methods.Photovoice is a well proven tool with the strength to convince people in charge, producing results based on robust evidence from practice and with the support of people in the situation being examined. Photovoice as a tool is useful to safety management since it can withstand criticism and resistance from those who protect their interests at the expense of health and safety and it helps to create joint ownership of a project outcome among participants and stakeholders.Using the photovoice approach helps to visualize and to provide underpinning of observable safety behaviour, safety culture and safety climate.Photovoice is suitable for many purposes in education and training in many areas, hence it seems likely that this also applies to health and safety.The photovoice method, applied in health and safety management, can open up new access windows to hitherto hidden and unknown situations, like it does in many other areas in society. Given its successful track record in many areas in society, we contend that photovoice has a high potential as a tool in support of health and safety management.”

The bullet points are appropriate for the six implementation steps. 

Ethical issues paragraph 5.3 need to be expanded, particularly the first paragraph.  I was unsure what some of the references such as “false light” were referring to. 

            We agree to this comment. The term “false light” is replaced by the generally accepted term  “framing”. We have rewritten and elaborated the first part of section 5.3:

“Several ethical issues need to be addressed in photovoice projects. The most fre-quently mentioned ethical issues with photovoice are: creation of unsafe conditions due to increased visibility of individuals, public exposure of participants, stigma, violations of confidentiality, invading privacy, involuntary intimacy, stereotyping, incriminating, framing, making profit on needy people, publishing of photographs of participants without their permission, disputing ownership of participants photographs, not ensuring informed consent, not taking into account illiteracy, not including an ensuring accurate expression of the group voice in reports [91,62,11,29,69,65,92,189,191,70,7,63,93,34,192,79].”

Again, it needs to be in narrative rather than bullet form. 

            We have included the improvement suggestions made by the reviewer. Both section 5.3 and 5.4 were changed into narrative format.

What were strategies the projects referenced used to protect their participants.  Put more plainly:  if I was an industrial or health worker, I would want to know exactly how presenting a photo of a safety violation is not going to lead to my termination.

            No explicit strategies were mentioned, only the need for participant protection was identified. To accommodate for this comment we added a sentence: ” Participants, e.g. workers, patients, need to know they are being protected against losing e.g. their job, reputation or social position.”

Limitations – there should be a few sentences more on your study limitations.

            We expanded section 5.5 Limitations with two sentences:

“This becomes clear from two extremes: merely taking photographs as data generated by an individual does not qualify as photovoice in our view.”  And:

“The other end of the range, in our view, is where the media generate huge numbers of static and moving images with explanatory text, all influencing the development of society on national and even global scale.

Future developments – Recommendations for choosing from the five generic designs is helpful for other researchers.  Overall, this paper is a good introduction for those considering photovoice as a research tool and is a meaningful contribution to the participatory research frameworks.

                Thank you for this positive comment.

Reviewer 2 Report

I have carefully read the manuscript titled "What about using photovoice for health & safety?" (IJERPH-1436384). The paper focuses on the possible applications and methods to include photovoice in health and safety procedures. I have found the paper interesting and well organized.

I can only suggest the Authors to revise the initial part of the text in order to make shorter some parts, such as the health promotion applications as well as the patient safety applications to improve the readibility of the text and have a more specific focus on the possible application of photovoice in the occupational health and safety system.  

Author Response

English language and style are fine/minor spell check required

            We have included improvement suggestions made by the reviewers.

I have carefully read the manuscript titled "What about using photovoice for health & safety?" (IJERPH-1436384). The paper focuses on the possible applications and methods to include photovoice in health and safety procedures. I have found the paper interesting and well organized.

                Thank you for this positive comment.

I can only suggest the Authors to revise the initial part of the text in order to make shorter some parts, such as the health promotion applications as well as the patient safety applications to improve the readibility of the text and have a more specific focus on the possible application of photovoice in the occupational health and safety system.   

 (See reviewer 1 comment)